# Pain Neuroscience Education and Physical Therapeutic Exercise for Patients with Chronic Spinal Pain in Spanish Physiotherapy Primary Care: A Pragmatic Randomized Controlled Trial

**DOI:** 10.3390/jcm9041201

**Published:** 2020-04-22

**Authors:** Miguel Angel Galan-Martin, Federico Montero-Cuadrado, Enrique Lluch-Girbes, María Carmen Coca-López, Agustín Mayo-Iscar, Antonio Cuesta-Vargas

**Affiliations:** 1Unit for Active Coping Strategies for Pain in Primary Care, East-Valladolid Primary Care Management, Castilla and León Public Health System (Sacyl), 47011 Valladolid, Spain; magalanm@saludcastillayleon.es (M.A.G.-M.); fmonteroc@saludcastillayleon.es (F.M.-C.); 2Doctoral Program of Research in Health Sciences, University of Valladolid, 47005 Valladolid, Spain; 3Department of Physical Therapy, University of Valencia, 46010 Valencia, Spain; 4Pain in Motion International Research Group, 1090 Brussels, Belgium; 5Department of Human Physiology and Rehabilitation Sciences, Faculty of Physiotherapy, Vrije University Brussels, B-1050 Brussels, Belgium; 6Castilla and León Regional Centre of Sports Medicine, (Sacyl), 47011 Valladolid, Spain; acuesta.var@gmail.com; 7Department of Statistics and Operational Research and IMUVA, University of Valladolid, 47005 Valladolid, Spain; 8Department of Physiotherapy, Faculty of Heath Sciences, University of Malaga, 19071 Málaga, Spain; 9Institute of Biomedical Research in Malaga. IBIMA, 29010 Málaga, Spain; 10School of Clinical Science, Faculty of Health Science, Queensland University Technology, Brisbane, QLD 4000, Australia

**Keywords:** chronic pain, chronic spinal pain, pain neuroscience education, physical exercise, primary care, randomized controlled trial

## Abstract

Chronic musculoskeletal pain affects more than 20% of the population, leading to high health care overload and huge spending. The prevalence is increasing and negatively affects both physical and mental health, being one of the leading causes of disability. The most common location is the spine. Most treatments used in the Public Health Services are passive (pharmacological and invasive) and do not comply with current clinical guidelines, which recommend treating pain in primary care (PC) with education and exercise as the first-line treatments. A randomized multicentre clinical trial has been carried out in 12 PC centres. The experimental group (EG) conducted a program of pain neuroscience education (6 sessions, 10 h) and group physical exercise with playful, dual-tasking, and socialization-promoting components (18 sessions in 6 weeks, 18 h), and the control group performed the usual physiotherapy care performed in PC. The experimental treatment improved quality of life (*d* = 1.8 in physical component summary), catastrophism (*d* = 1.7), kinesiophobia (*d* = 1.8), central sensitization (*d* = 1.4), disability (*d* = 1.4), pain intensity (*d* = 3.3), and pressure pain thresholds (*d* = 2). Differences between the groups (*p* < 0.001) were clinically relevant in favour of the EG. Improvements post-intervention (week 11) were maintained at six months. The experimental treatment generates high levels of satisfaction.

## 1. Introduction

Chronic musculoskeletal pain (CMP) affects more than one in five people [1,2,3], negatively affecting quality of life and generating suffering in people who suffer from it and in their companions [3,4]. The prevalence of CMP in the general population is increasing and has now become the leading cause of disability [5]. Furthermore, CMP also has a great economic impact related to the loss of working hours, disability, and the consumption of health resources [3].

Health systems are not giving a satisfactory response to patients with CMP, generating hopelessness and dissatisfaction both in patients and professionals [6]. The “Pain Proposal” initiative has shown that 85% of European doctors wish to receive additional information on pain management [6]. In addition, in the curricula of health sciences degrees, not enough training is received on how to manage patients with CMP [7,8]. Despite the recommendations of the current clinical evidence-based guidelines, which include patient education and physical exercise (PE) as first-line treatments [9,10,11,12,13,14,15], the most used treatments in these patients are pharmacological therapy [16,17] and surgery [18]. CMP is mistakenly treated as long-lasting acute pain, employing for this purpose all available therapeutic options, which has led to an alarming increase in the prescription of opioids. This situation has generated enormous concern both for the poor medium- and long-term effects achieved with opioid administration and the increase in side effects. In the USA, overdose deaths have increased fourfold in recent years [19,20,21,22]. This situation does not escape Spain (place of this clinical trial), where there has been an increase of more than 200% in the prescription of major opioids (fentanyl, tapentadol, and oxycodone), and in the regional Health Service where this trial was conducted, there are at moment more than 800 patients at risk for overdose [17].

In recent years, neuroscience has advanced in understanding pain mechanisms, including the role of central sensitization (CS) in CMP [23,24]. CS was originally defined as “an amplification of neural signalling within the central nervous system (CNS) that elicits pain hypersensitivity” [25] and is a broad concept encompassing numerous and complex pathophysiological mechanisms, such as a disrupted resting state functional connectivity in the default mode network, increased activity in nociceptive facilitatory pathways, or poor functioning of descending anti-nociceptive mechanisms [23,25,26,27]. Clinical criteria that allow the classification of patient pain as pain due to CS have been determined [28,29], as well as the various neuroplastic changes that occur in the CNS in the presence of CS [30,31,32]. In CS, there is an excessive response of CNS nociceptive neurons to both normal and subumbral stimuli, which is aggravated by an inactivation of endogenous pain inhibition mechanisms [33]. The presence of hyperalgesia, allodynia, and generalized pain in patients with CMP has been associated with the presence of CS [24]. Finally, patients with CS have been found to have high levels of kinesiophobia [34,35] catastrophism [34,36,37,38], fear-avoidance behaviours [38,39,40,41,42], and disability [2,3].

New therapeutic approaches based on pain neuroscience education (PNE) have shown positive effects on pain-related misconceptions, reducing catastrophism, kinesiophobia, and fear-avoidance behaviours [43,44,45,46,47,48,49,50]. There is a lot of variability in the intensity, content, and way of performing these interventions. Some research has shown that intense PNE interventions are more effective [51]. However, the effectiveness of PNE as an isolated treatment technique is limited [47] and its effects increase when combined with a targeted PE program [52,53]. PE is a first-choice treatment in the management of CMP [9,10,11,12,13,14], and PE in particular has proven to be a useful tool in the treatment of CMP. Furthermore, certain modalities of PE promote neurogenesis, neuroplastic changes at the brain level [54,55,56,57,58,59], and descending inhibitory pain pathways activation [60,61]. Thus, a therapeutic approach based on the combination of PNE and PE can improve quality of life and disability, decreasing the pain intensity of patients with CMP and those suffering from chronic spinal pain (CSP). Different clinical trials and reviews have shown promising results combining PNE and PE [45,46,48,62,63,64,65,66], but there is no consensus on the optimal dose of PNE or PE. These types of interventions, in which the patient is the active part of the treatment, are not performed and have been scarcely evaluated in primary care (PC).

Therefore, due to the increasing prevalence of CMP and the poor efficacy of most treatments applied to date, it is urgent to evaluate new therapeutic approaches in PC, which lack adverse effects and contribute to the sustainability of the health system. In this line, our group designed a new intervention [67], adapting the first-line recommendations of the current clinical evidence-based guidelines [9,10,11,12,13,14] to the PC setting. Looking for a consensus on the optimal doses of PNE and PE, with a standard and cost-effective intervention and in order to optimize the resources available, a “group treatment” program was chosen, as group treatment has previously been shown to be effective [68,69,70]. Regarding the educational intervention, an intense 10 h PNE program has been designed, while the PE group program was aimed at improving functionality, social interaction, physical contact, and progressively reaching levels close to 70% of maximum aerobic capacity through playful activities (gamification), double tasks, and challenges [67]. These components are intended to be tools that contribute to reversing the structural and functional changes characteristic of CS [30,71,72,73]. Patients with CSP were recruited, because this is the most prevalent location of CMP in the population [3,74]. Moreover, the mechanisms involved in the perpetuation of CMP are common regardless of the body region where the pain is located [75].

Thus, the objective of this study was to compare the effectiveness of a PNE and PE combination therapy program versus usual physiotherapeutic treatment used in PC physiotherapy units for CSP. It was hypothesized that the combined treatment of PNE and PE would be more effective in terms of quality of life, pain intensity, catastrophism, kinesiophobia, disability improvement, and CS-related variables than usual physiotherapeutic treatment in PC.

## 2. Experimental Section

A multicentric randomized clinical trial was conducted in 2017 in 12 PC centres belonging to the health areas of Valladolid-East and Valladolid-West of SACYL (Public Service of Health of Castilla and León, Spanish Public Health System). The trial was carried out in compliance with the recommendations of the SPIRIT [76] and CONSORT statements [77], and the design used the PRECIS-2 tool [78] (Appendix A, Figure A1). The study received favourable reports from the Ethical Committee for Clinical Research (CEIC) of Valladolid-East Health Area (CASVE-NM_16-252) and the CEIC of Valladolid-West Health Area (CEIC: 26/17). The trial was registered (Clinicaltrials.gov NTC03654235).

### 2.1. Patient Recruitment

Patients were recruited from the Family Medicine and Physiotherapy consultations of the participating health centres. Professionals involved in the recruitment process received a previous clinical session to learn about the study and to ensure that selected patients met the established inclusion criteria. Patient recruitment was carried out in January 2017. Patients aged between 18 and 70 years with non-specific CSP of more than 6 months duration were invited to participate in the study. The following conditions were exclusion criteria: cancer pain, fracture or surgical intervention of the spine in the last year, cognitive impairment that prevented the PNE program from being followed (in case of doubt, the Mini Mental Test was performed [79] with a minimum score of 25 [80]), physical condition that prevented the completion of the PE program (minimum requirement: execution in a normal time (<10″) of the Timed “Up and Go” Test [81]), pregnancy, cauda equina syndrome, chronic fatigue syndrome, fibromyalgia, complex regional pain syndrome, patients with associated pathologies that made it impossible to perform the PE program (myopathies and neurological disorders), and treatment with alternative therapies. The presence of pain in other anatomical regions, in addition to the spinal pain, was not a cause for exclusion. Selected patients were sent to the PC physiotherapy units where they received information about the clinical trial and signed the informed consent. Information provided to the participants sought to generate a neutral expectation related to the therapeutic procedures that were used in each study arm [82]. Study participants were not required to stop the analgesic drugs they were taking at the time of the study and were cited for initial evaluation with a blinded external evaluator.

### 2.2. Sample Size Calculation

A total of 63 individuals in each treatment group will allow the detection, in the physical component summary of SF-36, of differences between means of treatment groups in a magnitude corresponding to 50% of the standard deviation with a probability of 80%, keeping the type I error at a 5% level. By considering a possible attrition rate of 25%, it would be necessary to include 78 individuals in each group.

### 2.3. Randomization

Each patient evaluated was assigned an alphanumeric code. The alphabetical part of the code corresponded to the participant’s health centre, while the numerical part was allocated by correlative numbers. The randomization process was performed by the health technician of PC Valladolid-Este Management, who received a list of alphanumeric codes and made random assignments using the IBM SPSS 23 statistical package. Patients were assigned to the experimental group (EG) or control group (CG).

### 2.4. Blinding

Patient evaluations were conducted by evaluators who did not know the group to which the patient was assigned. The statistician who analysed the results was also blinded and did not know which intervention group each subject had been assigned to. Because of the characteristics of the intervention, neither the physiotherapists who performed it nor the patients who received it could be blinded. Physiotherapists who performed the interventions did not intervene in the process of assessing subjects.

### 2.5. Interventions

A detailed protocol of the methodology used has been published [67]. Physiotherapists participating in this study received prior training (30 h) about how to perform the studied interventions before the study commencement in order to standardize their practice and minimize uneven performance among therapists. All the physiotherapists received an instruction manual on how to carry out each of the program’s sessions in both arms. Two expert physiotherapists in the treatment of chronic spinal pain using coping strategies were responsible for the training of the physiotherapists who participated of this study. In addition, they all had a minimum experience of five years working with patients with chronic spinal pain in PC. Collaborators were instructed to record any incidence related to possible adverse effects that may be related to the interventions.

#### 2.5.1. Experimental Group Intervention

Patients assigned to this group performed a PNE program consisting of six sessions (10 h) and 18 sessions of therapeutic PE to be performed in six weeks (18 sessions/18 h) with a frequency of three sessions per week. The program was performed in the order shown in Figure 1.

PNE was carried out in a group in each health centre and was taught in all centres by the same physiotherapists. The contents of the educational program and the way in which it was carried out are described in the study protocol [67].

PE was carried out in a group. During the sessions, repeated reference was made to the theoretical content learned in the first part of the program (PNE), with the aim of making patients understand the reason for each component introduced in the PE protocol. Precautions were taken to adapt PE dosing to the functional state of patients [83]. The fact that the intervention was in a group was not an obstacle for each patient to receive individualized indications on how to perform or adapt the exercises to their own state. Because patients with CSP may have ineffectiveness of descending inhibitory pain pathways [84], the first sessions could lead to an increase in symptomatology. Patients were warned of this possibility and at no time was the onset of pain reason to stop the activity [85,86].

The intensity of exercise was progressively increased through the use of games, double tasks, challenges, and elements that stimulated physical contact and social interaction as described in the study protocol [67]. At all times, during the PE sessions, the achievement of objectives was prioritized through playful tasks over the quality of the execution of movements.

#### 2.5.2. Control Group Intervention

The CG received the usual physiotherapy treatment that is carried out in the PC physiotherapy units, and at the time the trial was carried out, it was supported by the current protocols of physiotherapy in PC of Castilla and León [87]. Treatment consisted of 15 sessions (15 h) of thermotherapy and analgesic electrotherapy in the area or areas of pain, and exercises recommended by the Spanish Society of Physical Medicine and Rehabilitation (SERMEF) were prescribed [88]. The learning and execution of the exercises was supervised by a PC physiotherapist.

### 2.6. Outcome Variables

The study participants were evaluated before the intervention, after the intervention (week 11), and at 6 months (week 26). Evaluators were PC physiotherapists from health centres who did not participate in the study and had a minimum of 5 years of experience in the health system.

#### 2.6.1. Personal and Sociodemographic Variables

The initial pre-intervention evaluation recorded age, gender, health centre, level of studies, employment status, marital status, anthropometric variables (weight, size, body mass index, and abdominal perimeter), and duration of symptoms (months).

#### 2.6.2. Primary Outcome Variables

The main measure of outcome was the difference between groups in the change in health-related quality of life (HRQL) at different times (initial assessment, post-intervention, and 6 months). The Spanish version of the SF-36 v2 health survey was used. SF-36 measures HRQL in both the general population and specific subgroups, has very good psychometric qualities, and measures eight health and wellness variables, as well as a summary of mental and physical health based on psychometry. Each variable is transformed directly on a scale from 0 to 100. Higher scores indicate an overall improvement in quality of life. Both the physical and mental summary components were adjusted by age and sex to the population reference values [89,90,91].

#### 2.6.3. Secondary Outcome Variables

“Adherence to treatment” was measured by recording the number of sessions performed in each of the arms. The results were expressed in percentages.

“Catastrophism” was measured using the pain catastrophizing scale (PCS) [92]. The PCS is a very sensitive scale for evaluating pain-related behaviours and cognition [93]. The score ranges from 0 to 52. Higher scores indicate higher levels of catastrophism.

“Kinesiophobia” was measured using the Spanish version of the Tampa Scale of Kinesiophobia (TSK-11) [94]. This self-reported questionnaire consists of 11 items that value the patient’s fear of moving and re-injuring. Scores range from 11 to 44 points. Higher scores indicate higher levels of kinesiophobia.

“CS” was measured using the Spanish version of the central sensitization inventory (CSI) [95]. This questionnaire provides highly reliable and valid data to quantify the severity of various symptoms presented by patients with CS [96], and it is considered the best self-informed tool to value CS [29]. High CSI scores are related to high intensities of pain, anxiety, depressive symptoms, symptoms of somatization, disability, and sleep disorders [97]. Scores in part A of the questionnaire catalogue the severity levels of CS (0–29: subclinical; 30–39: mild; 40–49: moderate; 50–59: severe; and 60–100 extreme) [98].

“Disability” was assessed through the Roland–Morris disability questionnaire [99]. This self-reported disability questionnaire consists of 24 questions related to physical functions that may be altered by back pain. The score range is 0–24. Higher scores indicate higher levels of disability.

McGill’s pain maps were used to measure the location and extent of pain areas [100]. For the intensity of pain, the analogue visual scale (VAS) was used with an unmarked 100 mm line [101], and algometry was used to determine pressure pain thresholds (PPTs). PPTs were determined using the protocol described by Neziri et al. [102] applied at four reference points (midpoint between a horizontal line drawn between the side edge of the acromion and the spinal process of the seventh cervical vertebral, bilaterally, and the midpoint between the highest part of the superior border of the iliac crest and the spinal process at the same height, also bilaterally). An algometer with a rubber tip with an application area of 1 cm^2^ was used (Warner Instruments FPX-100). The average of two consecutive measurements was determined with a minimum interval of 30 min between each measurement [103].

“Analgesic intake” was registered before the intervention and at six months.

The Client Satisfaction Questionnaire (CSQ-8) was used in the post-intervention assessment to assess the satisfaction of the participants with the treatment received [104]. The CSQ-8 is a self-report tool used to assess satisfaction with health services.

### 2.7. Statistical Analysis

Numerical variables were summarized with means ± standard deviations and 95% confidence intervals (CI 95%). For summarized categorical variables, we calculated percentages. Distributions of numerical variables were checked if they had a normal distribution. The only numerical variable showing evidence of belonging to a non-normal distribution was months with pain, which was summarized with quartiles (Q1, Q2, and Q3). We used box and whisker plots for representing numerical variables. For comparing changes in means inside groups and between groups, we applied the paired and unpaired Student’s t-test, respectively. *p*-values <0.05 were considered statistically significant. In relation to effect size interpretation (Cohen’s *d*), we proposed the following: negligible (0–0.19), small (0.2–0.49), moderate (0.5–0.79), and large (>0.8). Data analysis was performed by using the R statistical package.

## 3. Results

A total of 205 CSP patients were recruited at PC consultations during the second half of January 2017. Figure 2 represents the flowchart for this study.

The drop-out rate during follow-up was 5.6% in EG vs. 9.9% in CG at post-intervention evaluation, and 8.9% in EG vs. 14.9 in CG at six months.

Regarding adherence to treatment, in the EG, 94% performed more than 80% of PNE sessions and 79.2% more than 80% of PE sessions. In the CG, 88.9% performed all the sessions and 92% performed more than 80%.

No adverse effect was recorded in any of the groups.

With the enrolled individuals in both groups, the power of the carried study for detecting differences, at the level considered as relevant in the given sample size calculation, was 86%, greater than the 80% initially planned.

The characteristics of the recruited sample are reflected in Table 1, with characteristics of the recruited sample showing sociodemographic data of the recruited population.

### Primary and Secondary Outcomes

The HRQL outcomes (SF-36) and the rest of the secondary outcomes can be seen in Table 2. Graphical representation of secondary outcomes can be seen in Figure 3.

In health-related quality of life, there were differences between groups (*p* < 0.001) in favour of the EG. Post-intervention changes were maintained at 6 months. Effect size in the EG was large in the physical component summary and moderate in the mental component summary.

In the secondary variables, there were differences between the groups (*p* < 0.001) in favour of the EG. Effect size in the CG was large for the decrease of catastrophism, kinesiophobia, CSI score, disability, pain intensity, and increase of PPT. Changes made after the intervention were maintained at 6 months.

Analgesics consumption was similar in both groups at the initial evaluation: 92% in EG vs. 89% in CG (*p* = 0.64). In the evaluation at six months, analgesics consumption was lower in EG group (46%) than in EG group (78%) (*p* < 0.001).

Satisfaction for the service received was higher in the EG than in the CG, highlighting the high scores obtained in quality of performance (Excellent: 93% EG vs. 43% CG), satisfaction with the help received (Very satisfied: 93% EG vs. 28% CG), and program recommendation (Yes definitely: 96% EG vs. 40% CG).

## 4. Discussion

In this study, combined treatment with PNE and PE was compared with usual physiotherapy care in patients with CSP who access PC physiotherapy units of the Spanish Public Health System (Community of Castilla and León). The outcomes have shown that a combined PNE and PE program was more effective than the conventional treatment in all studied variables, both at the end of the treatment and at 6 month follow-up. In addition, participants’ satisfaction with the treatment received was higher in the EG as compared with that in the group that received usual physiotherapy care.

The EG achieved a large effect size on quality of life improvement in all dimensions of the SF-36 physical health-related questionnaire, highlighting improvement in “body pain” and “general health” subscales and approaching the reference values of the Spanish population [90]. Similar effects were obtained in the EG in the subsections “social function” and “vitality”, while in “mental health” and “emotional health”, the effect size was moderate. These results are in agreement with a recent study that compared the application of education combined with a goal setting intervention for patients, which obtained a large effect size in improvement of quality of life, although smaller than that obtained by our EG [53]. Similar results in terms of the physical summary component were achieved in an intervention based in education and a multimodal physical therapy program with deep-water running, although in that study, outcomes were measured at 15 weeks and patients were younger than those in our study [64]. Another recent study found moderate effect size after the application of PNE combined with cognition-targeted motor control training in patients with CSP [62]. We consider that an intervention with an extensive PNE program and PE with playful components is responsible for this improvement in post-intervention quality of life that is maintained at 6 months.

As we expected, combined PNE and PE was more effective than conventional physiotherapy treatment in terms of reduction of catastrophism. A recent trial, which was conducted outside the public health system, also managed to have a large effect on reducing catastrophism [105], but in this case at 3 months. We have seen that, at 6 months, the reduction was sustained in our EG. Systematic reviews of interventions that have combined PNE with different modalities of physiotherapy have shown clinically relevant reductions in catastrophism, but less than those found in our EG [46,47]. The program that has been designed for the EG, with intensive educational intervention, could explain these results.

Reduction in kinesiophobia was also much greater in the EG. It is known that cognitive changes derived from educational intervention produce changes in the quality and quantity of movement [49]. Other studies, like us, have found a large effect size in the reduction of kinesiophobia when PNE and PE are combined [105], which is in line with what recent revisions point to [46,48]. PNE in isolation also produces clinically significant improvements in catastrophism and kinesiophobia, despite having a small effect size in disability [47].

Reductions in CSI scores were very important in the EG. Approximately three-quarters of patients achieved scores that placed them at subclinical levels (below 30) [98]. There are few studies with which we can compare this effect. A Belgian study achieved significant reductions in CSI scores both at 6 months and 1 year [62]. Reductions obtained in our study at 6 months were slightly higher, with a large effect size. We believe that this difference may be due to a longer-term educational intervention that may have helped more to change wrong beliefs, and it has been combined with PE that includes a progressive increase in intensity thanks to group play activities, which also includes components that facilitate physical contact and social interaction.

Disability reduction has been clinically very relevant in the EG and in line with the findings of recent reviews combining PNE with other forms of physiotherapy and PE [46,48]. The effect size was large and was similar to that obtained with other similar interventions published in a recent study [105].

Reduction in pain intensity, as well as number of body areas with pain, was higher in the EG compared to the CG. This was an expected finding, as there was a significant decrease in pain intensity in persistent pain processes related to CS (fibromyalgia) in previous studies in which high doses of PNE were applied [51]. In subjects with chronic lower back pain, a significant reduction in pain intensity was also observed after application of PNE and PE in combination [46,47,48,105,106]. In our EG, effect size was large and greater than the referenced studies. Furthermore, a significant increase in PPT in the EG was achieved with large effect sizes and in line with what has been achieved in other studies [105].

It is worth highlighting that small effect sizes were obtained in the group of subjects who received the usual physiotherapy treatment. However, as other studies have shown, the existence of CS predicts poor outcomes obtained with classic local treatments, such as electrotherapy, motor control exercises, and surgery [107,108]. Sometimes PE can increase symptoms because descending inhibitory systems are not working properly [83], and some authors even point out that exercise can induce hyperalgesia if psychosocial factors such as fear of pain, catastrophism, and erroneous beliefs are not previously addressed, and this was not done in the CG [109]. Neither classical physiotherapy intervention nor drug treatment achieved clinically significant improvements. This situation generates a continuous pilgrimage through different specialties and an overuse of health resources, and it helps to consolidate the belief that CMP is for life and there are no solutions to this condition [110]. These poor clinical outcomes with passive techniques could generate despair in the patient, negatively influence his/her mood, and lead to a catastrophic situation accompanied by kinesiophobia, greater pain and disability, and consequently a worse quality of life. While regular physiotherapy interventions based on analgesic electrotherapy and exercise, such as those received by the CG, may have some usefulness in acute and subacute musculoskeletal pain [12]. Subjects with SC-associated pain will potentially respond better to interventions aimed at desensitizing their CNS, such as the intervention received by the EG [111]. In addition to complying with the recommendations of the clinical practice guidelines, it also followed the eleven recommendations that appeared in the review of Lin et al. [14].

In an attempt to adapt the intervention of the EG to PC, we chose to include a series of components in the treatment program considered of crucial importance for the outcomes obtained. The first one was to choose an intense educational intervention similar to that used in previous studies [51], which might have favoured the modification of beliefs about pain [112], and that was continually strengthened throughout the program with the delivery of support material [113,114,115]. The second important aspect was the fact that the intervention of the EG was applied in a group, which favoured socialization [68,69,70] and took many patients out of their social isolation. Another important component was the inclusion of games in PE, which had a powerful distracting effect and positively influenced the modification of movement patterns, allowing subjects to overcome kinesiophobia and to achieve levels of aerobic training without being aware of the intensity of the effort made, with the consequent benefits of PE at this intensity. The use of distracting elements, such as the dual-task activities, during PE have also been important, as well as the challenges to which they have been continuously subjected [116,117,118]. In short, the experimental treatment program has managed to empower the patients by providing them with tools that make them the active part of the treatment.

Despite the good outcomes obtained, about one-fifth of the EG patients, at the end of the program and at 6 months, had levels of pain and disability that limited their functionality. For these patients, who have not been so good responders, we must consider other therapeutic alternatives, such as the collaboration of psychology professionals, application of pharmacological treatment, and individualized intensive physiotherapy.

### 4.1. Practical Implications and Recommendations for Research

In view of the results obtained in this study, the authors considered that patients with CSP should receive the health care they need in PC. The experimental intervention consisting of PNE and PE proved to be more effective than conventional physiotherapy treatments, requiring few resources, lacking side effects, and producing significant clinical improvements in these patients. It has to be a priority of health managers to enhance therapeutic approaches that empower the patient, provide him/her with up-to-date knowledge about pain neuroscience and provide him/her with tools that improve functionality and decrease disability. A change in the CMP approach model that is more focused on biopsychosocial aspects and avoids the use of passive or drug-only treatments is needed. It is essential to bet on the education and empowerment of clinicians and patients and to provide tools to promote the self-management of the patient [119].

It would be interesting to assess the longer-term effects of the intervention, as well as to assess how to act in the event of a relapse with these patients. In this line, our working group is compiling the clinical results of hundreds of patients who are carrying out the program. This activity is also allowing us to collect numerous data that will allow us to develop clinical prediction rules to identify good responders. Data are also being collected on the effect of active coping programs on drug use to be published soon. Finally, we need to check whether the intervention has produced changes in brain connectivity, which is why our group is also working with brain imaging experts.

### 4.2. Limitations and Strengths

The main limitation of this study was that the intervention carried out by the CG lacked educational intervention and was based on protocols that, although they were in force in our health system, are from 2005 [87]. Burden of intervention differences between groups (in favour of experimental group) may have influenced the results of this study due to factors related to the therapist–patient relationship. Even so, the treatment received by the CG is the usual treatment that is used in the vast majority of PC units of the Spanish health system and was taken as the best possible comparator due to the fact that they are supported by protocols in force in the health service where the study was conducted.

In this study, mainly female patients with diffuse and nonspecific pain profiles were recruited, although it corresponds to the profile of the patient who consults the most for pain on PC, it could be that the results of our study cannot be extrapolated to the general population.

Patients in both arms continued to take the analgesic medication they were prescribed, and although taking analgesic drugs in CSP is not a first-line recommendation [9,10,11,12,13,14,15], the influence of the treatments received on the analgesics consumption requires a larger and more focused study in this regard.

Regarding the strengths of the study, it should be noted that the intervention was carried out within usual health care and had a marked pragmatic component. As a result, the transfer of the intervention to the clinical reality can be done easily. It should also be noted the sample size used, as far as the knowledge of the authors reaches, is to date the largest clinical trial conducted in this line of work.

## 5. Conclusions

An intervention based on PNE and PE with playful, dual-tasking, and socialization-promoting components is most effective in terms of quality of life improvement, reduction of pain, catastrophism, kinesiophobia, CS, and disability, and it generates high levels of satisfaction compared to the usual physiotherapy treatment in subjects with CSP treated in PC physiotherapy units.

## Figures and Tables

**Figure 1 jcm-09-01201-f001:**
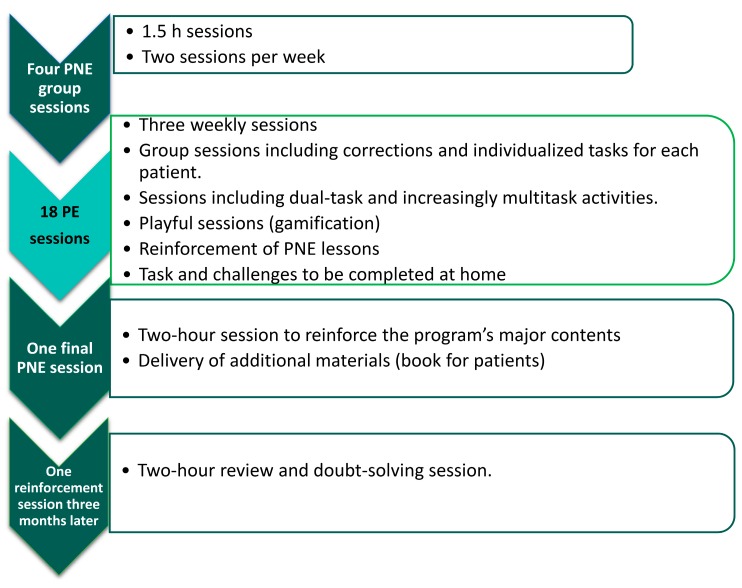
Distribution of the program followed by the experimental group.

**Figure 2 jcm-09-01201-f002:**
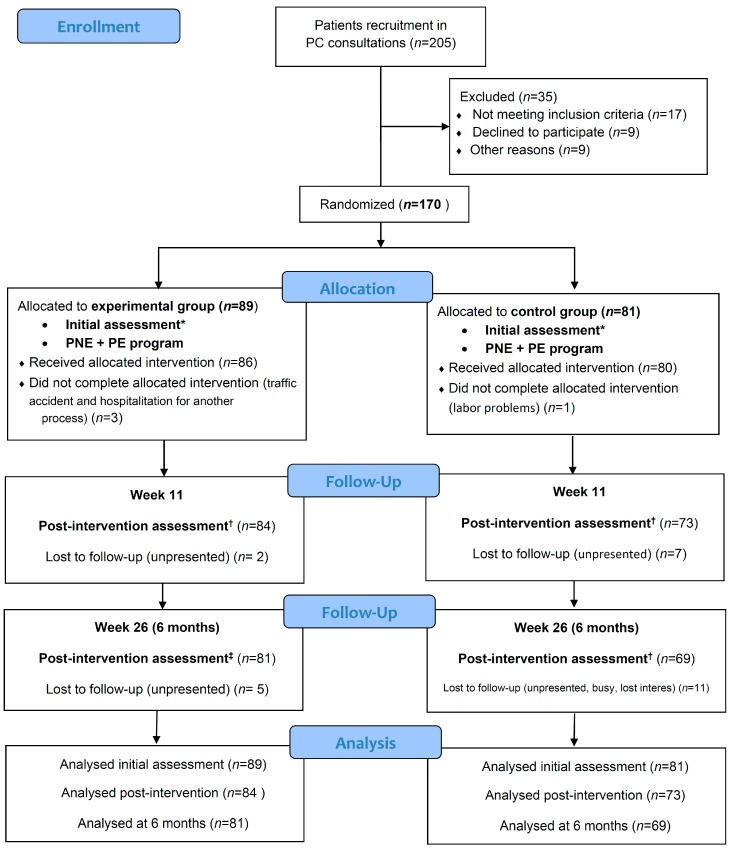
CONSORT flow diagram. * Initial assessment included initial personal and sociodemographic variables, health-related quality of life (SF-36), catastrophism (PCS: pain catastrophizing scale), kinesiophobia (TSK-11: Tampa Scale for Kinesiophobia-11), central sensitization (CSI: central sensitization inventory), disability (RMDQ: Roland–Morris disability questionnaire), pain intensity (VAS: visual analogue scale), pain areas (McGill maps), and algometry (PPTs: pain pressure thresholds). ^†^ Post-intervention assessment satisfaction survey with healthcare received (CSQ-8: client satisfaction questionnaire-8) was added. ^‡^ The 6-month assessment was the same as the initial assessment.

**Figure 3 jcm-09-01201-f003:**
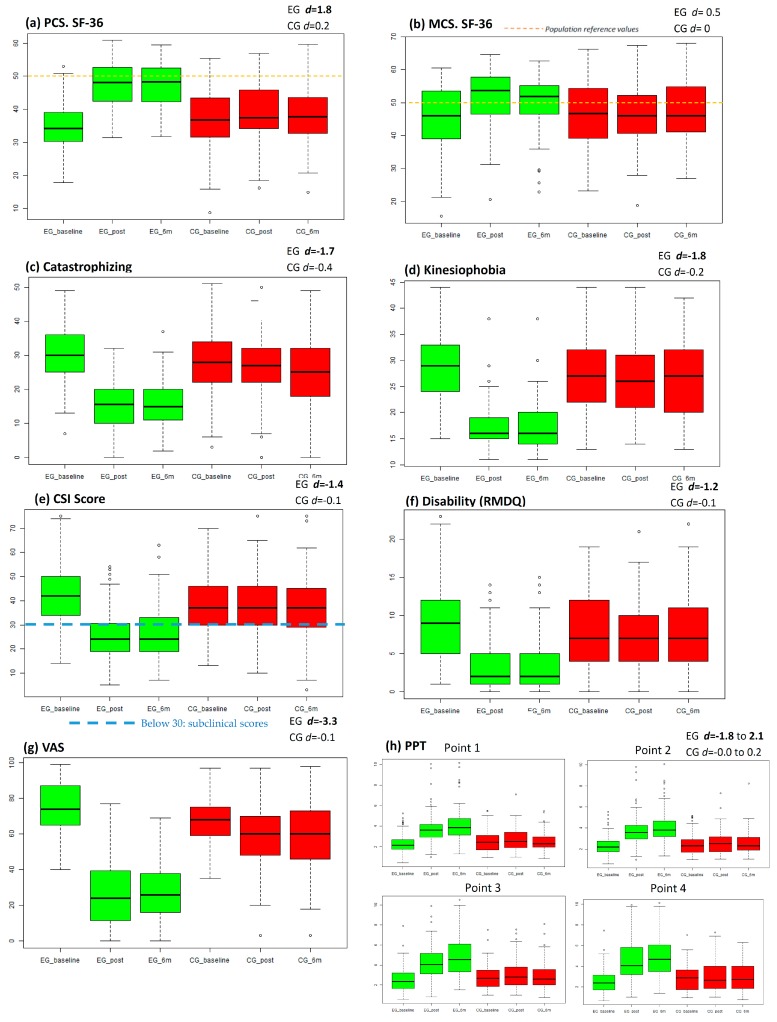
Outcome graphical representation. This figure contains outcomes graphical representation in boxplot. Green colour represent experimental group (EG) and red colour represent control group (CG). First boxplot represents the initial assessment, the second boxplot represents post-intervention assessment and the third the assessment at six months; (**A**) Boxplot for physical component summary (PCS); (**B**) Boxplot for mental component summary (MCS); (**C**) Boxplot for catastrophism outcomes; (**D**) Boxplot for kinesiophobia outcomes (TSK-11); (**E**) Boxplot for CSI scores (part A); (**F**) Boxplot for disability (RMDQ); (**G**) Boxplot for intensity of pain (VAS; (**H**) boxplot for PPT outcomes; EG *d* = X.X: Experimental Group effect size at six months (Cohen’s *d*); CG *d* = X.X: Control group effect size at six months. Hollow circles represents outlier values.

**Table 1 jcm-09-01201-t001:** Characteristics of the recruited sample.

	EG (*n* = 89)	CG (*n* = 81)
Age (mean ± SD)	53.02 ± 10.7	49.14 ± 12.14
Female (n)	65	71
**Marital Status (%)**		
*Married/in couple*	62	57
*Divorced*	5	5
*Single*	14	15
*Sidow*	8	4
**Education Level (%)**		
*Incomplete Primary educat.*	3.7	6.2
*Primary education*	33.7	35.8
*Lower secondary education*	16.9	17.3
*Upper secondary education*	12.4	7.4
*Post-secondary non-tertiary*	14.6	17.3
*University degree*	19.1	16.0
**Employment Situation**		
*Student*	0.0	2.5
*Active*	40.4	51.9
*Unemployed*	22.5	12.3
*Home chores*	15.7	19.8
*Pensioner*	21.3	13.6
**Anthropometrics**		
BMI (mean ± SD)	27.6 ± 4.7	26.5 ± 5.0
Pain		
*VAS (mm) (mean±SD)*	70.8 ± 14.8	67.2 ± 14.3
*Months with pain (mean±SD)*	93.13 ± 83.5	93.86 ± 84.91
*Mouths with pain (median)*	48	48
CSP (%)	100	100
Cervical pain	80.9	79.3
Thoracic pain	65.2	50.0
Low back pain	91.0	89.0
Pain in Other Areas (%)		
RUL	42.7	35.4
LUL	40.4	31.7
RLL	37.1	34.1
LLL	25.8	23.2

EG: experimental group; CG: control group; BMI: body mass index; Q1: first quartile; Q2: median; Q3: third quartile; CSP: Chronic spinal pain; VAS: visual analogue scale RUL: Right upper limb; LUL: Left upper limb; RLL: Right lower limb right; LLL: left lower limb.

**Table 2 jcm-09-01201-t002:** Primary and secondary outcomes.

	Group EG (*n* = 89) CG (*n* = 81)	Pre	Post	6 mo	Intra. Dif (6 mo – Pre) 95% IC	Effect Size	Dif. Inter 6 mo (EG – CG) 95% IC
**HRQL (SF-36)**							
Physical Function (PF)	EG	60 ± 18.2	80.2 ± 13.4	81.4 ± 13.8	22 (18.9, 25.2) **	1.2	18.7 (14.1, 23.4) ^†^
	CG	63.1 ± 23.3	65.8 ± 22.3	65.8 ± 20.9	3.3 (−0.1, 6.7)	0.1	
Role Physical (RP)	EG	47.5 ± 26.3	77.8 ± 20.7	78.2 ± 20.4	32.3 (26.4, 38.2) **	1.2	30.7 (21.6, 39.8) ^†^
	CG	56.2 ± 25.7	58.1 ± 25.4	59.1 ±24.7	1.6 (−5.2, 8.5)	0.1	
Bodily Pain (BP)	EG	32.7 ± 17.1	75.8 ± 16.5	73.1 ± 14.9	41 (36.9, 45) **	2.4	34.9 (28.3, 41.3) ^†^
	CG	36.7 ± 19	44.2 ± 20.8	42.8 ± 23.2	6.1 (1, 11.2) *	0.3	
Social Functioning (SF)	EG	60.5 ± 24.1	88.1 ± 15.4	84.3 ± 18.5	24.1 (18.9, 29.3) **	1	22.7 (14.3, 30.9) ^†^
	CG	68.7 ± 24.6	71.6 ± 25.3	70.7 ± 24.7	1.4 (−5.1, 8)	0.1	
Mental Heath (MH)	EG	61.4 ± 18.6	77.6 ± 15.7	74.1 ± 17.2	12.8 (9.3, 16.3) **	0.7	12.4 (7, 18) ^†^
	CG	63.3 ± 19.5	63.4 ± 17.3	64.4 ± 19	0.4 (−3.9, 4.6)	0	
Role Emotion (RE)	EG	74.2 ± 24.7	89.2 ± 15.4	90.3 ± 15.1	16.9 (11.7, 22) **	0.7	16.7 (9.3, 23.9) ^†^
	CG	80.2 ± 21.8	80.6 ± 21.4	82.6 ± 20.1	0.2 (−4.8, 5.3)	0	
Vitality (VT)	EG	40 ± 18.4	65.1 ± 17.8	59.8 ± 18.5	20.3 (16.3, 24.3) **	1.1	17.4 (10.7, 24.1) ^†^
	CG	42.8 ± 20.1	44.1 ± 18.4	46.8 ± 19.6	2.9 (−2.5, 8.3)	0.1	
General Health (GH)	EG	42.4 ± 16.7	66.2 ± 16.8	65.4 ± 17.7	24 (20.4, 27.6) **	1.4	23.5 (18.1, 28.9) ^†^
	CG	48.9 ± 19.9	50.7 ± 18.2	48.6 ± 20.4	0.5 (−3.5, 4.5)	0	
Health Transition	EG	30.1 ± 18.1	76.2 ± 15	75.9 ± 14.5	45.4 (40.5, 50.2) **	2.5	37.1 (29, 45.1) ^†^
	CG	36.1 ± 19.4	41.4 ± 22.1	43.8 ± 23.6	8.3 (1.9, 14.8)	0.4	
**Physical component summary (PCS)**	EG	34.9 ± 7.3	47.4 ± 6.9	47.4 ± 6.7	12.9 (11.4, 14.3) **	1.8	11.4 (9.1, 13.6) ^†^
	CG	37.1 ± 9.3	38.9 ± 9	38.3 ± 8.5	1.5 (−0.2, 3.3)	0.2	
**Mental component summary (MCS)**	EG	44.9 ± 10.1	51.7 ± 7.7	50 ± 7.9	5.1 (3.2, 7.1) **	0.5	5.2 (2.2, 8.1) ^†^
	CG	46.6 ± 10.2	46.5 ± 8.8	47.3 ± 9.3	−0.1 (−2.3, 2.1)	0	
**Catasthrophism (PCS)**	EG	30.3 ± 8.7	15.4 ± 7.3	15.5 ± 7.2	−15 (−16.7, −13.4) **	−1.7	−11 (−13.6, −8.4) ^†^
	CG	27.9 ± 9.1	26.6 ± 9.7	24.2 ± 10.3	−4 (−6, −2.1) **	−0.4	
Rumination	EG	10.5 ± 2.8	5.6 ± 2.6	5.5 ± 2.6	−5 (−5.6, −4.5) **	−1.8	−3.6 (−4.5, −2.7) ^†^
	CG	9.6 ± 3.1	9.2 ± 3.2	8.2 ± 3.6	−1.4 (−2.1, −0.7) **	−0.5	
Magnification	EG	6.3 ± 2.3	3.1 ± 2	3.2 ± 2	−3.2 (−3.7, −2.7) **	−1.4	−2.5 (−3.3, −1.6) ^†^
	CG	5.8 ± 2.7	5.5 ± 2.8	5.3 ± 2.6	−0.7 (−1.4, −0.1) *	−0.3	
Helplessness	EG	13.5 ± 4.6	6.7 ± 3.7	6.7 ± 3.8	−6.8 (−7.8, −5.9) **	−1.5	−4.9 (−6.4, −3.6) ^†^
	CG	12.5 ± 4.6	11.9 ± 4.8	10.7 ± 5	−1.9 (−2.8, −0.9) **	−0.4	
**Kinesiophobia (TSK-11)**	EG	28.9 ± 6.6	17.1 ± 4	17.2 ± 4.7	−12.2 (−13.5, −10.9) **	−1.8	−10.6 (−12.4, −8.7) ^†^
	CG	27.5 ± 7.1	26.1 ± 6.3	26.3 ± 7.6	−1.6 (−3, −0.3) *	−0.2	
**Central Sensitization (CSI)**	EG	43.4 ± 12.5	25.7 ± 10.8	25.8 ± 10.5	−17.7 (−19.3, −16) **	−1.4	−16.6 (−19, −14.1) ^†^
	CG	38.6 ± 11.7	37.7±12.4	37.4 ± 13.5	−1.1 (−2.9, 0.6)	−0.1	
Disability (RMDC)	EG	9.2 ± 4.8	3.3 ± 3.5	3.3 ± 3.8	−6 (−6.8, −5.2) **	−1.2	−5.6 (−6.7, −4.5) ^†^
	CG	8 ± 4.7	7.6 ± 4.6	7.7 ± 4.8	−0.4 (−1.1, 0.2)	−0.1	
Pain intensity (VAS)	EG	74.1 ± 14.5	26.7 ± 18	27 ± 16.2	−48.2 (−52.6, −43.8) **	−3.3	−40.9 (−46.7, −35.2) ^†^
	CG	67.2 ± 14.3	58.4 ± 17.7	59.7 ± 19.8	−7.3 (−10.6, −3.9) **	−0.5	
Algometry (PPT). P1	EG	2.4 ± 1	3.8 ± 1.5	4.2 ± 1.7	1.8 (1.5, 2.2) **	1.8	1.8 (1.4, 2.2) ^†^
	CG	2.5 ± 1	2.7 ± 1.1	2.5 ± 0.9	0 (−0.2, 0.3)	0	
Algometry (PPT). P2	EG	2.3 ± 0.9	3.8 ± 1.5	4.1 ± 1.7	1.9 (1.6, 2.2) **	2.1	1.7 (1.3, 2.2) ^†^
	CG	2.5 ± 1	2.6 ± 1.1	2.6 ± 1.1	0.2 (−0.1, 0.5)	0.2	
Algometry (PPT). P3	EG	2.5 ± 1.2	4.3 ± 1.6	4.9 ± 2	2.4 (2, 2.8) **	2	2.2 (1.7, 2.8) ^†^
	CG	2.8 ± 1.2	3.1 ± 1.5	3 ± 1.4	0.2 (−0.1, 0.5)	0.2	
Algometry (PPT). P4	EG	2.6 ± 1.2	4.5 ± 1.8	5 ± 2.1	2.5 (2, 2.9) **	2.1	2.3 (1.7, 2.9) ^†^
	CG	2.8 ± 1.3	3.1 ± 1.5	3 ± 1.4	0.2(-0.1,0.5)	0.2	

Pre: Values before intervention; Post: Values post-intervention (week 11); 6 mo: Values six months; Intra. Dif: Intragroup difference; Inter. Dif.: Intergroup difference; * and ** indicates significant intragroup differences between six months assessment and initial assessment (*p* < 0.05 and *p* < 0.001, respectively); **^†^** indicates significant intergroup differences between six months assessment and initial assessment (*p* < 0.001).

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
