# Peer review of "Pain Neuroscience Education and Physical Therapeutic Exercise for Patients with Chronic Spinal Pain in Spanish Physiotherapy Primary Care: A Pragmatic Randomized Controlled Trial"

_jcm, 2020, doi:10.3390/jcm9041201_

Round 1

Reviewer 1 Report

Thank you for the opportunity to read this comprehensive and highly planned RCT study.

Information on treatment procedures, including coping strategies, is needed. This study fits into that context.

1
Introduction: it is informative but very comprehensive. With reference to the study protocol, it can advantageously be focused and shortened.

Line 62: The 800 patients are out ...?

Experimental Section:

2.1

All patients are recruited over 2 weeks in January - which is a very short period to recruit more than 150 patients. Do you think it affects generalizability?

Most of the patients are women, and it seems that the unemployed are over-represented in the intervention group - is that something which has to be considered?

2.2

The effect of the treatment is defined as 50% of the standard deviation between pairs. What is the primary outcome of the calculation and why is the clinically important difference not used?

line 162 the numbering is confusing is Intervention 2.5?

3 Results

Nothing is stated in accordance to compliance. Did the patients attended all sessions (except for the few mentioned in the flowchart), and what about the control group, what type of treatment did they actually receive?

4. Discussion

I am very surprised that there is no effect in the control group especially since the treatment in this group includes both passive treatment methods and instruction in exercises.
Normally, we see an improvement in the short term just as patients have been in contact with therapists.
This is expected to be discussed in more detail in the discussion.

Author Response

ITEMIZED LIST OF THE REVIEWERS’ COMMENTS

Journal of Clinical Medicine

(jcm-762843)

Dear Editor,

Please, find a revision of our manuscript entitled “Pain neuroscience education and physical therapeutic exercise for patients with chronic spinal pain in Spanish physiotherapy primary care: A pragmatic randomized controlled trial”.

First of all, we would like to thank the Reviewers for their their thoughtful and constructive comments and the opportunity to revise this paper. We have considered all your suggestions, and have incorporated them into the revised manuscript. Changes to the original manuscript have been highlighted with “track changes” mode (in yellow background). We believe our manuscript is stronger as a result of the modifications. In addition, an itemized point-by-point response to the Reviewers’ comments is presented below. 

REVIEWER 1

Thank you for the opportunity to read this comprehensive and highly planned RCT study.

Information on treatment procedures, including coping strategies, is needed. This study fits into that context.

1
Introduction: it is informative but very comprehensive. With reference to the study protocol, it can advantageously be focused and shortened.

Line 62: The 800 patients are out ...?

Authors:

Dear reviewer. First of all, we wanted to thank the time invested by you to review the document. We are sure that your comments and corrections will favor a development in the quality and understanding of the document.  Accordingly, we have shortened the introduction section as suggested. The 800 patients mentioned in the manuscript belong to the Health Service where the trial was performed.  We gave this data to emphasize the importance of chronic spinal pain within the National Health System but that’s not the number of subjects participating in the study. In order to avoid confusion, we have changed that part of the manuscript. It now reads: “This situation does not escape Spain (place of this clinical trial), where there has been an increase of more than 200% in the prescription of major opioids (Fentanyl, Tanpentadol and Oxycodone), and  in the regional Health Service where this trial was conducted, there are at moment more than 800 patients at risk for overdose [17]”  

Experimental Section:

2.1

All patients are recruited over 2 weeks in January - which is a very short period to recruit more than 150 patients. Do you think it affects generalizability?

Most of the patients are women, and it seems that the unemployed are over-represented in the intervention group - is that something which has to be considered?

Authors:

We understand your concern regarding recruitment issues here. Indeed, recruitment was probably the easiest part of this study as twelve Primary Care centres with 100 GPs participated in the recruitment process, so it was possible to be completed in two weeks. We think our sample represents the type of patients who visit Primary Care centres looking for relief of their chronic spinal pain. Regardless, an exhaustive description of the type of patients who normally attend Primary Care centres for treating their chronic spinal pain is currently in preparation. In this study, we will describe how gender or unemployment can affect patients.

2.2

The effect of the treatment is defined as 50% of the standard deviation between pairs. What is the primary outcome of the calculation and why is the clinically important difference not used?

line 162 the numbering is confusing is Intervention 2.5?

Authors:

Thank you for alerting us of this important issue. The primary outcome of the calculation is the effect size in Physical Component Summary of SF-36 that includes, among other variables, bodily pain. Accordingly, we have rewritten the description of the sample size calculation for a better comprehension. This has now been rephrased to: A total of 63 individuals in each treatment group will allow detecting, in Physical Component Summary of SF-36, differences between means of treatment groups in a magnitude corresponding to 50% of the standard deviation with a probability of 80%, keeping the type I error at a 5% level. By considering a possible attrition rate of 25%, it would be necessary to include 78 individuals in each group.”

The minimal clinically important difference has not been used because according to the consulted papers, the MCID data for Chronic spinal pain differed greatly between different authors. We have also changed the numbering of the intervention.

3 Results

Nothing is stated in accordance to compliance. Did the patients attended all sessions (except for the few mentioned in the flowchart), and what about the control group, what type of treatment did they actually receive?

Authors:

Thank you for your comment. We proceed to incorporate the results of adherence of each of the arms. It now reads: Adherence to treatment was measured by recording the number of sessions performed in each of the arms. The results were expressed in percentages”

  1. Discussion

I am very surprised that there is no effect in the control group especially since the treatment in this group includes both passive treatment methods and instruction in exercises.
Normally, we see an improvement in the short term just as patients have been in contact with therapists.
This is expected to be discussed in more detail in the discussion.

Authors:

Thank you for pointing this out. In patients with chronic pain associated with high levels of central sensitization, several studies have found that they respond poorly to passive therapies and we have indicated this in the discussion (ref. 107).  We have added more references to this in the discussion (108). In this type of patients, physical exercise is an effective therapeutic option when it is associated with educational intervention (which was not received by the control group). Other authors point out that EF may have an effect contrary to that sought by not properly functioning the descending inhibitory system, in this line other works point out that exercise-induced hyperalgesia can be generated if psychosocial factors such as fear of pain, kinesiophobia, catastrophism and wrong beliefs. We proceed to modify and reference this topic in the discussion. It now reads “It is worth highlighting that small effect sizes were obtained in the group of subjects who received the usual physiotherapy treatment. But, as other studies have shown, the existence of CS predicts poor outcomes obtained with classic local treatments, such as electrotherapy, motor control exercises and surgery [107,108]. Sometimes PE can increase symptoms because descending inhibitory systems are not working properly [83], and some authors even point out that exercise can induce hyperalgesia if psychosocial factors such as fear of pain, catastrophism and erroneous beliefs are not previously addressed, and this was not done in the CG [109]. Neither classical physiotherapy intervention nor drug treatment achieved clinically significant improvements

REVIEWER 2

Dear authors,

 In this manuscript, the authors conducted a pragmatic, multi-centered, assessor- and analyst-blinded, randomized controlled clinical trial which compared the effectiveness of a PNE and PE combination therapy program versus usual physiotherapeutic treatment used in primary care physiotherapy units for chronic spinal pain (CSP) in Spain. The study seemed generally well designed with good ethical issues, especially in compliance with the recommendation of the SPIRIT and CONSORT statements. However, there are some points being discussed or being improved with balanced view points.

Major points

  1. One of issue is “Is there any difference of the use of analgesic drugs during the trial between groups?”. Analgesic drugs may affect the pain outcomes.

Authors:

Dear reviewer, thank you very much for the time spent reviewing this document, for your comments and suggestions.

Authors:

Indeed, there was not between-groups difference in the analgesic drugs intake at initial assessment. We have added some new information in the limitations to make this point clear. It now reads: “Patients in both arms continued to take the analgesic medication they were prescribed, and although taking analgesic drugs in CSP is not a first-line recommendation [9-15], the consumption of drugs could have negatively or positively influenced the results.”.

The moderate to large effect size might be come from the difference of the dosage of PNE and PE combination versus usual physiotherapy treatment. This is closely related to the frequency of contact (18 sessions vs. 15 sessions), longer time for treatment, and closer contact to participants.

Authors:

Thanks for alerting us of this important issue.  We admit that dosage differences may have influenced the results of this study and explain between-groups differences in treatment response. In order to account for this issue we have now added a new sentence/paragraph in the limitations section. It now reads: Burden of intervention differences between groups (in favour of experimental group) may have influenced the results of this study due to factors related to therapist-patient relationship. However, given the pragmatic nature of the trial, experimental treatment was compared to usual care where 15 treatment sessions are generally performed. The experimental group receive 18 sessions because we consider that it is the minimum time necessary for the intervention to be effective.

  1. Another issue is 2.2. sample size calculation in page 3 to 4.

The explanation of sample size calculation seems very ambiguous. Please supplement the description of sample size calculation be more comprehensive. I wonder you adopted 25% drop-outs, but the re-calculation considering 25% drops makes 84 (63/0.75 = 84), but not 78.

Authors:

Thank you very much for your comment. Accordingly, we have rewritten the description of the sample size calculation for a better comprehension. This has now been rephrased to: A total of 63 individuals in each treatment group will allow detecting, in Physical Component Summary of SF-36, differences between means of treatment groups in a magnitude corresponding to 50% of the standard deviation with a probability of 80%, keeping the type I error at a 5% level. By considering a possible attrition rate of 25%, it would be necessary to include 78 individuals in each group.”

The usual formula that we applied for obtaining the sample size provided us 63 inviduals by arm for getting the desired power. After it, we calculated the 25% of this quantity, by assuming it as an upper bound for the losses, and we added it to the initial one for getting the final sample size. 

  1. Secondary outcomes should include safety measure. Please report safety issues related to two different interventions. You mentioned “The experimental intervention consisting of PNE and PE proved to be more effective than conventional physiotherapy treatments, requires few resources, lacks side effects and produces significant clinical improvements in these patients.” In page 13, but you might not be able to say ‘lacks side effects’ because of not evaluating the safety outcome.

Authors:

Thank you for this valuable comment. Indeed adverse treatment effects were recorded in both groups and were non-existent in either group. We have added new information in the manuscript related to safety issues. In 5.2. it now reads: Collaborators were instructed to record any incidence related to possible adverse effects that may be related to the interventions”.

  1. Is there any program for the standardization of the practice of physiotherapists? How did you modulate the uneven performance among therapists? Is there any minimum requirement of career for physiotherapists involved in the trial?

Authors:

Thank you for these questions. Indeed, all the physiotherapists participating in this study received prior training (30 hours) about how to perform the studied interventions before the study commencement in order to standardize their practice and minimize uneven performance among therapists. All the Physiotherapist  received an instruction manual on how to carry out each of the program's sessions in both arms. Two physiotherapists expert in the treatment of chronic spinal pain using coping strategies were responsible of the training of the physiotherapists who participated of this study.  In addition, all the physiotherapists had a minimum experience of five years working with patients with chronic spinal pain in Primary Care.

All this information has been now added in the manuscript. It now reads: “Physiotherapists participating in this study received prior training (30 hours) about how to perform the studied interventions before the study commencement in order to standardize their practice and minimize uneven performance among therapists. All the Physiotherapist received an instruction manual on how to carry out each of the program's sessions in both arms. Two physiotherapists expert in the treatment of chronic spinal pain using coping strategies were responsible of the training of the physiotherapists who participated of this study.  In addition, they all had a minimum experience of five years working with patients with chronic spinal pain in PC.

  1. Please supplement 4.2. Limitations and strengths in page 13 with balanced view points. In my opinion, the trial is apt to be biased because experimental group is more chance to contact, higher dosage of treatment, and non-blinding of practitioners and participants may lead to positive effectiveness. On the contrary the study is well pragmatically designed, powered sample sized, randomized trial seems strengths in real world setting.

Authors:

Thank you, new information is added in 4.2. It now reads “Burden of intervention differences between groups (in favour of experimental group) may have influenced the results of this study due to factors related to therapist-patient relationship.

Minor points

  1. There were many abbreviations in the manuscript made the readers confused. Therefore, if the lists of abbreviations were given in the manuscript, that will be very easy to read the manuscript.

Authors:

Thank you for your comment. We have made a list of abbreviations in the manuscript, we have inserted before the reference list.

  1. Please add more important keywords in page 1 as ‘pragmatic, randomized controlled trial’.

Authors:

Done, thank you.

  1. The sentence of “In recent years, neuroscience has advanced in understanding pain mechanisms, ~~ fear-avoidance behaviours [41–45], and disability [2,3]” seemed to relate to central sensitization (CS). However it seemed not to be main topic but you want to illustrate the relationship between chronic pain and central sensitization. Please be short or remain related context within the mainstream.

Authors:

Thanks for your comment, we have reduced that paragraph substantially. (line 65)

  1. Please present drop-out rate of your study.

Authors:

Drop-out rates have been added in Results. Adherence to treatment has been also added in results, expressing the percentage of sessions performed.

Authors:

REVIEWER 3

R3 his study aimed to validate, in an RCT, the efficacy of a rehabilitation program combining pain neuroscience education and physical activity. The control group received what is called an “usual physiotherapy care” which consist of 15 sessions of thermotherapy and analgesic electrotherapy in the area of pain, and the guidelines for the exercises recommended by the Spanish Society of Physic Medicine and Rehabilitation. The experimental protocol has already been published elsewhere.

The results obtained in the present study are very promising as the experimental group displayed a systematic improvement of all outcome variables investigated. Interestingly quite significant effects have been found on improvement in QOL and pain management.

Comments:

The study has been well conducted in terms of methodology. However, there are few limitations that need to be addressed by the authors.

As discussed by the authors, a main limitation of the study comes from the fact that the control group received a treatment that is very different, in quality and quantity, when compared to the EG. The latter seems to be much more intensive that in the CG. It could be worth to give more information about the total duration of the physical therapies in the two groups and to better describe the CQ intervention.

Authors:

Thank you very much for the time spent reviewing the manuscript and for your interesting suggestions. Accordingly, we have added new information in the limitations section to account for between-group differences in treatment in terms of dosage and therapist-patient relationship and how these factors may have influenced the results of this study. It reads: Burden of intervention differences between groups (in favour of experimental group) may have influenced the results of this study due to factors related to therapist-patient relationship”. In addition, we have also add more details about the total duration of the physical therapies used un both groups (line 195 and line 217), although this information was already included in the referenced published protocol.

Another limitation that need to be addressed is the specificity of the population. This study focused mainly on female patients with non-specific and diffuse pain profiles. This may reduce generalization of the results and need to be discussed.

Authors:

Thank you for alerting us of this important issue. We recognize the lack of generalization of our results to the general population and have added new information in the limitations section to account for it. It reads “In this study, mainly female patients with diffuse and nonspecific pain profiles were recruited, although it corresponds to the profile of the patient who consults the most for pain on PC, it could be that the results of our study cannot be extrapolated to the general population". We are preparing a descriptive study of the type of users with CSP treated in Primary Care. We have found that women are more frequent and that the percentage of women who go to PC is higher than that shown by the prevalence surveys of persistent pain. We modify the wording for the record.

Comparison of the medication between the 2 groups should be performed as the data is available. It could be interesting to also address evolution of medication intake after the intervention (if available).

Authors:

Thank you for your comment.  

Authors:

We have added some new information in the limitations to make this point clear. It now reads: “Patients in both arms continued to take the analgesic medication they were prescribed, and although taking analgesic drugs in  is CSP not a first-line recommendation [9-15], the consumption of drugs could have negatively or positively influenced the results.”.

One fifth of the patients from the EG did not respond to the treatment. This need to be discussed. It could be interesting to determine if these patients differ somehow from the general population.

Authors:

Thank you for this comment. One of the most interesting aspects of our line of work is that we are developing clinical prediction rules that allow us to identify good and bad responders for this type of therapeutic approach. Those interesting rules will be the subject of a future paper.

Please explain why authors focused on QOL as main outcome instead of pain interference, pain intensity or disability, for instance.

Authors:

That’s a very interesting point. Our main outcome was QOL because we have planned in a second part of this study to assess the QUALYs provided by the experimental intervention. This will in turn allow us to investigate in a third study the potential of implementing a therapeutic program such as the one received by the experimental group in Primary Care centers.

Line 74 : What is “subumbral “ stimuli?

Authors:

Subumbral stimulus is a stimulus that under normal conditions would not generate a pain response, but pain would appear in situations where the nervous system is sensitized.

Thank you. We reiterate the previous words. Thanks to your effort in reviewing the document, we believe that it has improved. 

Reviewer 2 Report

Dear authors,

In this manuscript, the authors conducted a pragmatic, multi-centered, assessor- and analyst-blinded, randomized controlled clinical trial which compared the effectiveness of a PNE and PE combination therapy program versus usual physiotherapeutic treatment used in primary care physiotherapy units for chronic spinal pain (CSP) in Spain. The study seemed generally well designed with good ethical issues, especially in compliance with the recommendation of the SPIRIT and CONSORT statements. However, there are some points being discussed or being improved with balanced view points.

Major points

  1. One of issue is “Is there any difference of the use of analgesic drugs during the trial between groups?”. Analgesic drugs may affect the pain outcomes.
  2. The moderate to large effect size might be come from the difference of the dosage of PNE and PE combination versus usual physiotherapy treatment. This is closely related to the frequency of contact (18 sessions vs. 15 sessions), longer time for treatment, and closer contact to participants.
  3. Another issue is 2.2. sample size calculation in page 3 to 4.

The explanation of sample size calculation seems very ambiguous. Please supplement the description of sample size calculation be more comprehensive. I wonder you adopted 25% drop-outs, but the re-calculation considering 25% drops makes 84 (63/0.75 = 84), but not 78.

  1. Secondary outcomes should include safety measure. Please report safety issues related to two different interventions. You mentioned “The experimental intervention consisting of PNE and PE proved to be more effective than conventional physiotherapy treatments, requires few resources, lacks side effects and produces significant clinical improvements in these patients.” In page 13, but you might not be able to say ‘lacks side effects’ because of not evaluating the safety outcome.
  2. Is there any program for the standardization of the practice of physiotherapists? How did you modulate the uneven performance among therapists? Is there any minimum requirement of career for physiotherapists involved in the trial?
  3. Please supplement 4.2. Limitations and strengths in page 13 with balanced view points. In my opinion, the trial is apt to be biased because experimental group is more chance to contact, higher dosage of treatment, and non-blinding of practitioners and participants may lead to positive effectiveness. On the contrary the study is well pragmatically designed, powered sample sized, randomized trial seems strengths in real world setting.

Minor points

  1. There were many abbreviations in the manuscript made the readers confused. Therefore, if the lists of abbreviations were given in the manuscript, that will be very easy to read the manuscript.
  2. Please add more important keywords in page 1 as ‘pragmatic, randomized controlled trial’.
  3. The sentence of “In recent years, neuroscience has advanced in understanding pain mechanisms, ~~ fear-avoidance behaviours [41–45], and disability [2,3]” seemed to relate to central sensitization (CS). However it seemed not to be main topic but you want to illustrate the relationship between chronic pain and central sensitization. Please be short or remain related context within the mainstream.
  4. Please present drop-out rate of your study.

Thank you,

Author Response

(The authors gave the same response as above.)

Reviewer 3 Report

Aim

This study aimed to validate, in an RCT, the efficacy of a rehabilitation program combining pain neuroscience education and physical activity. The control group received what is called an “usual physiotherapy care” which consist of 15 sessions of thermotherapy and analgesic electrotherapy in the area of pain, and the guidelines for the exercises recommended by the Spanish Society of Physic Medicine and Rehabilitation. The experimental protocol has already been published elsewhere.

The results obtained in the present study are very promising as the experimental group displayed a systematic improvement of all outcome variables investigated. Interestingly quite significant effects have been found on improvement in QOL and pain management.

Comments:

The study has been well conducted in terms of methodology. However, there are few limitations that need to be addressed by the authors.

As discussed by the authors, a main limitation of the study comes from the fact that the control group received a treatment that is very different, in quality and quantity, when compared to the EG. The latter seems to be much more intensive that in the CG. It could be worth to give more information about the total duration of the physical therapies in the two groups and to better describe the CQ intervention.

Another limitation that need to be addressed is the specificity of the population. This study focused mainly on female patients with non-specific and diffuse pain profiles. This may reduce generalization of the results and need to be discussed.

Comparison of the medication between the 2 groups should be performed as the data is available. It could be interesting to also address evolution of medication intake after the intervention (if available).

One fifth of the patients from the EG did not respond to the treatment. This need to be discussed. It could be interesting to determine if these patients differs somehow from the general population.

Please explain why authors focused on QOL as main outcome instead of pain interference, pain intensity or disability, for instance.

Line 74 : What is “subumbral “ stimuli?

Author Response

(The authors gave the same response as above.)

Round 2

Reviewer 1 Report

The response is adequate and my concerns are met.

Author Response

ITEMIZED LIST OF THE REVIEWERS’ COMMENTS

Journal of Clinical Medicine

(jcm-762843)

Dear Editor,

Please, find a revision of our manuscript entitled “Pain neuroscience education and physical therapeutic exercise for patients with chronic spinal pain in Spanish physiotherapy primary care: A pragmatic randomized controlled trial”.

We would like to thank the reviewers for their comments and proposed changes, as we believe that they have contributed significantly to the improvement of the document.

R1

The response is adequate and my concerns are met.

Authors:

Thank you again for your valuable contributions that have contributed to a significant improvement of the manuscript.

R2

Firstly, thank you very much for your efforts to response to the reviewers’ comments. After the revision, the manuscript seemed to be much improved.

  1. For solving the issue of sample size estimation, please calculate the power of your results (by primary outcome) for preventing the study from under-powered issue. If you present the study’s power, you can prove your study is powered one.

Authors:

Thank you for your comment and for your interesting suggestion, we have calculated the statistical power a posteriori. You can see it in the results section.

(line 271): “With the enrolled individuals in both groups, the power of the carried study for detecting differences, at the level considered as relevant in the given sample size calculation, was 86%, greater than the 80% initially planned.”

  1. Another issue is I recommend you to present the use, or consumption of analgesic drug between the groups at the end of treatment/ or follow-up. The answer of “the consumption of drugs could have negatively or positively influenced the results” seemed very ambiguous. If the intervention of PNE and PE be very effective, it has a possibility to be able to reduce opioid consumption.

Authors:

Thanks for the comment. We have added analgesics intake data.

(line 247): Analgesic intake was registered before the intervention and at six months.”

 (line 299): “Analgesics consumption were similar in both groups at the initial evaluation: 92% in EG vs 89% in CG (p=0.64). In the evaluation at six months, analgesics consumption was lower in EG group (46%) than in EG group (78%) (p <0.001).

We have changed a phrase in the discussion that, as you point out, was very ambiguous. Now it reads: the influence of the treatments received on the analgesics consumption requires a larger and more focused study in this regard.”

 We consider that the document has improved qualitatively with these modifications.

  1. It is very strange that “No adverse effects were recorded in any of the groups” is very uncommon conditions or under-estimated conditions compared to normal clincal studies.

Authors:

In therapeutic physical exercise interventions supervised by physical therapists, adverse effects are unlikely in patients with chronic pain [1] . Changes in pain intensity are common during sessions or during program performance below 20%, but due to the type of patients, these fluctuations have not been considered relevant [2]. There were no adverse effects understood as an unwanted experience during the follow-up that led to contact with the health system (general practitioner or hospital). There were also no serious adverse effects that required surgery, caused permanent disability, or death [3]

  1. Geneen, L.; Smith, B.; Clarke, C.; Martin, D.; Colvin, L.A.; Moore, R.A. Physical activity and exercise for chronic pain in adults: an overview of Cochrane reviews. Cochrane Database Syst. Rev. 2017, 4.
  2. van Tulder, M.; Malmivaara, A.; Hayden, J.; Koes, B. Statistical Significance Versus Clinical Importance. Spine (Phila. Pa. 1976). 2007, 32, 1785–1790.
  3. FDA What is a Serious Adverse Event? Available online: https://www.fda.gov/safety/reporting-serious-problems-fda/what-serious-adverse-event.

We greatly appreciate your comments on this topic. Thank you for your important contribution.

R3

I would like to thank the authors for their answers and the adaptations performed on the manuscript. The latter being now, in my opinion, fully suitable for publication. However, it could be worth to go through the text once again, as I noticed 1-2 typo errors.

Thank you very much, the document has been revised again. We have found and corrected the two typo errors.

Reviewer 2 Report

Dear authors,

Firstly, thank you very much for your efforts to response to the reviewers’ comments. After the revision, the manuscript seemed to be much improved.

  1. For solving the issue of sample size estimation, please calculate the power of your results (by primary outcome) for preventing the study from under-powered issue. If you present the study’s power, you can prove your study is powered one.
  2. Another issue is I recommend you to present the use, or consumption of analgesic drug between the groups at the end of treatment/ or follow-up. The answer of “the consumption of drugs could have negatively or positively influenced the results” seemed very ambiguous. If the intervention of PNE and PE be very effective, it has a possibility to be able to reduce opioid consumption.
  3. It is very strange that “No adverse effects were recorded in any of the groups” is very uncommon conditions or under-estimated conditions compared to normal clincal studies.

Thank you

Author Response

(The authors gave the same response as above.)

Reviewer 3 Report

I would like to thank the authors for their answers and the adaptations performed on the manuscript. The latter being now, in my opinion, fully suitable for publication. However, it could be worth to go through the text once again, as I noticed 1-2 typo errors.

Author Response

(The authors gave the same response as above.)
